# PERM1—An Emerging Transcriptional Regulator of Mitochondrial Biogenesis: A Systematic Review

**DOI:** 10.3390/genes15101305

**Published:** 2024-10-09

**Authors:** Eveline Soares Menezes, Zeyu Wu, John R. M. Renwick, Andres Moran-MacDonald, Brendon J. Gurd

**Affiliations:** 1School of Kinesiology and Health Studies, Queen’s University, Kingston, ON K7L3N6, Canada; 19esm2@queensu.ca (E.S.M.); cary.wu@queensu.ca (Z.W.); 17aamm1@queensu.ca (A.M.-M.); 2Department of Kinesiology, McMaster University, Hamilton, ON L8S 4K1, Canada; renwickj@mcmaster.ca

**Keywords:** oxidative metabolism, endurance exercise, cardiac function, therapeutic target, skeletal muscle adaptation, disease models

## Abstract

Background/Objectives: This systematic review aims to explore the role of PERM1 across different organisms, tissues, and cellular functions, with a particular focus on its involvement in regulating skeletal muscle mitochondrial biogenesis. Methods: This systematic review follows The PRISMA 2020 Statement. We used the Covidence systematic review software for abstract/title screening, full-text review, and data extraction. The review included studies that examined PERM1 expression or activity in skeletal muscle, heart, and adipose tissue and/or cells, from mice, rats, and humans, and involved exercise or disease models. Risk of bias was assessed using the Cochrane Collaboration tool, and the data were extracted and synthesized qualitatively, with bioinformatic analyses performed using the MetaMEx database. Results: Twenty-one studies were included in our data extraction process, where 10 studies involved humans, 21 involved mice, four involved rats, and 11 involved cells. Conclusions: PERM1 in skeletal muscle increases with endurance exercise, affecting muscle function and oxidative metabolism, but its role in humans is not well understood. In cardiac tissue, PERM1 is vital for function and mitochondrial biogenesis purposes, but decreases with disease and pressure overload. Our review synthesizes the current understanding of PERM1’s function, raises awareness of its role in mitochondrial regulation, and identifies key areas for future research in the field.

## 1. Introduction

Mitochondrial biogenesis—the making of new components of the mitochondrial reticulum [1]—initiates the integration of newly generated protein into pre-existing mitochondria, leading to improvements in mitochondrial content and function [1,2,3,4]. Much of the current understanding of the molecular control of mitochondrial biogenesis centers around PGC-1α, the purported “master regulator of mitochondrial biogenesis” [5,6]. With respect to the skeletal muscle, research has focused on PGC-1α because of its transcriptional activation by contraction-sensitive pathways, including AMPK, CaMK, p38 MAPK, and SIRT1 [3]. However, although muscle PGC-1α is indispensable for basal levels of mitochondrial content and enzymatic/respiratory function [5,6,7], it is dispensable for the adaptive response to exercise training [8,9,10]. The dispensability of PGC-1α has stimulated interest in alternative/novel regulators of transcription and has led to the emerging understanding that mitochondrial biogenesis is controlled by a complex and redundant regulatory network [1].

One emerging regulator of skeletal muscle mitochondrial biogenesis is PPARGC1- and ESRR-induced regulator in muscle 1 (PERM1). PERM1 is highly expressed in skeletal muscle and heart tissue and is involved in several cellular processes including the contractile function, muscle mitochondrial biogenesis and oxidative capacity, glucose and lipid metabolism, and energy transfer [11,12,13]. Although the work of Cho and colleagues has received attention by several recent reviews [14,15,16], a comprehensive review of PERM1 function in different tissues and model organisms is currently unavailable. Thus, the aim of this systematic review is to comprehensively examine the role of PERM1 in various organisms, tissues, and cellular functions with a focus on its putative role in the regulation of skeletal muscle mitochondrial biogenesis. Our review synthesizes what is currently understood about PERM1 function, raises awareness about one of the less conventional genes/proteins involved in the regulation of mitochondrial biogenesis, and highlights important future directions for research on PERM1 and the field of mitochondrial biogenesis.

## 2. Materials and Methods

This systematic review adhered to The PRISMA 2020 Statement [17] (see the Appendix A). Abstract/title screening, full-text review, and data extraction were conducted using the Covidence systematic review software (https://www.covidence.org/, accessed on 1 October 2024). The selected methods were registered on Open Science Framework (https://osf.io/ksqzp/) on 23 February 2023, prior to the initial literature search, data extraction, and data analysis.

### 2.1. Eligibility Criteria

Studies were included if they: (1) characterized PERM1 expression/activity in any type of tissue, cell, or organism, (2) examined cells, mice, and humans of all ages under any condition (e.g., rest vs. exercise, inactivity, diseases, physical activity vs. inactivity, etc.), (3) used PERM1 transgenic mice, PERM1 infected and/or knockdown cells, (4) used exercise (acute and training protocols) and/or disease models to investigate PERM1, and (5) were published in English or could be translated effectively using Google translate. Studies were excluded if they: (1) did not meet all inclusion criteria and (2) could not be sourced from two different libraries or after repeated online searches and an attempt to contact the authors.

### 2.2. Search Strategy

We conducted a literature search of studies indexed in the PubMed, Web of Science, SCOPUS, Ovid (MEDLINE, EMBASE, AMED, APA, Psychinfo, Cochrane), EBSCO (CINAHL, SPORTDiscus), and Pedro databases on 23 February 2023, following protocol registration (https://osf.io/ksqzp/). The literature search using the same databases was updated on 30 August 2024 to ensure recently published studies were included. Search strategies included subject headings and key words related to PERM1 function, expression, and activity (see the Appendix A). After the MEDLINE strategy was finalized, we adapted it to the subject headings of the other databases. PROSPERO was also searched for ongoing or recently completed systematic reviews. Titles and abstracts were then imported into Covidence from the database searches, and duplicates were removed by Covidence automatically. Studies that were not peer-reviewed or only had abstracts available were also excluded from Covidence. The electronic database search was supplemented by including any publications known to the authors that were not found in the electronic databases. To ensure literature saturation, we examined the reference lists of the studies included or of relevant reviews identified through the search. We also searched the authors’ personal files to make sure that all relevant material was captured.

Two independent reviewers evaluated each article during both the screening and the data extraction phases using the Covidence software. In an attempt to ensure consistency in the review process, all reviewers were familiarized to our inclusion criteria prior to the start of screening/extraction, and discussions were held throughout the review process as questions/conflicts arose. During the data extraction phase, reviewers utilized a data extraction template generated on Covidence.

Any conflicts that arose during the screening and extraction processes were resolved through discussion between the two reviewers who initially screened/extracted the data from a given article. If consensus could not be reached through discussion, a third reviewer was consulted to resolve the conflict.

### 2.3. Selection Process

Two review authors independently screened each title/abstract against the eligibility criteria. Full reports were obtained for all titles that appeared to meet the eligibility criteria or for any study where there were some uncertainties. Two review authors then screened the full text reports and decided whether the inclusion criteria were met. If there was a disagreement between the two authors on whether to include a study, a third author screened the text and made a decision with respect to the study inclusion. Additional information was sought out from the study authors when there were questions about eligibility.

### 2.4. Risk of Bias Assessment

The Cochrane Collaboration Risk of Bias Assessment Tool [18] was used to assess the risk of bias for each study, as we have done previously [19] (See the Appendix A for the results). This tool covers sequence generation, allocation concealment, blinding, incomplete outcome data, and selective outcome reporting. A judgement was made with respect to the possible risk of bias on each of the five domains, with risk being classified as either “high risk” or “low risk”. If there were insufficient details reported in the study, the risk of bias was reported as “unclear”. These judgements were made independently by two of the review authors, with disagreements being resolved by discussion and, when necessary, by consultation with a third reviewer.

### 2.5. Data Extraction and Data Synthesis

Results were extracted and synthesized by the lead author, with a qualitative summary of the extracted data being presented in the final review. There was no meta-analysis conducted in this study; instead, data were extracted (see the Appendix A) and then discussed in a narrative fashion.

A meta-analysis was considered during the planning stage of this review. However, we ultimately decided that a meta-analytical approach was not warranted given the high heterogeneity in study design/methodology, species/tissue type, and outcome measures across available studies. Thus, to prevent a misleading and/or underpowered meta-analysis, a narrative synthesis was selected as a more appropriate method to comprehensively summarize the findings across studies.

### 2.6. Bioinformatic Analyses

We conducted a bioinformatic analysis to examine changes in PERM1 in a large dataset of physical inactivity and acute and training aerobic, resistance, HIIE, and combined studies. These analyses were completed using MetaMEx, a gene expression database containing meta-analyses of skeletal muscle response to exercise https://www.metamex.eu (accessed on 22 July 2024) [20]. Using MetaMEx, we customized the search criteria to define the human population of interest. The search parameters were specified as follows: (i) for sex, we included studies involving males, females, and those with undefined sex, (ii) for age, we included studies involving young, middle-aged, and elderly individuals, (iii) for fitness, we included sedentary, active, and athletic individuals, (iv) weight categories encompassed lean, overweight, obesity, and obesity class 3, (v) muscle groups included vastus lateralis, biceps brachii, soleus, and gastrocnemius, and (vi) for health status, we selected studies involving only healthy individuals. All available timepoints (immediate, 1 h, 3 h, 4 h, 5 h, 6 h, 8 h,18 h, 24 h, 48 h, and 96 h post exercise) were included.

## 3. Results

The study selection process is summarized in Figure 1. We identified a total of 71 studies through our comprehensive database searches. After removing 24 duplicates, we screened the titles and abstracts of the remaining 47 studies. Twenty studies were selected for full-text screening. One additional study (human) was included as a cross-referenced study that was not captured in our database search but that met the inclusion criteria, bringing the total number of included studies to 21. All 21 studies were included in our data extraction process.

The distribution of the studies by species and tissue type is illustrated in Figure 2. Our findings revealed that 10 studies involved humans, 21 involved mice, four involved rats, and 11 involved cells. The distribution of human studies was as follows: five focused on skeletal muscle, three on the heart, and two on other tissues. For the mouse studies, eight focused on skeletal muscle, nine on the heart, and three on other tissues. The rat studies included two on muscles, one on the heart, and one on other tissues. Cell line studies involved four in C2C12 (skeletal muscle) cells, four in cardiomyocytes, and three on other cell types (HEK-293, adipocytes, and blood cells). A summary of all studies, including model, intervention, and key findings for PERM1-related outcomes in skeletal muscle tissue, heart tissue, and other tissues are presented in Table 1, Table 2 and Table 3.

### 3.1. Risk of Bias

Full details for our risk of bias analysis are presented in the Appendix A. We observed high rates of high/unclear risk of bias for selection bias (sequence generation, 95% high/unclear; allocation concealment, 86% high/unclear), performance bias (95% high/unclear), detection bias (95% high/unclear), attrition bias (86% high/unclear), and reporting bias (100% unclear). None of the studies were judged to have a low risk for all sources of bias. Thirteen out of 21 studies were judged to have an unclear risk for all sources of bias. Only one study reported sample size calculations.

### 3.2. MetaMEx

Results from our bioinformatics analyses using MetaMEx—including all available timepoints—are presented in Figure 3 and in Appendix A.

## 4. Discussion

This systematic review examined the function of PERM1 in various organisms, tissues, and cells. A total of 21 studies were included that reported analysis performed in different species (human, mice, rats, cells) across a variety of tissues, including muscle, heart, tumor, brain, and adipose tissues. Our findings reveal that PERM1 (1) is a transcriptional regulator highly expressed in oxidative muscle fibers and heart tissue, (2) is inducible by exercise, (3) is involved in the control of skeletal muscle mitochondrial biogenesis, and (4) coordinates the energetics and function of heart tissue via mitochondrial function, substrate metabolism, and lipids and amino acid homeostasis.

### 4.1. PERM1 and Skeletal Muscle

We were able to find 13 studies examining PERM1 in skeletal muscle (summarized in Table 1). These studies demonstrate that PERM1 is highly expressed in skeletal muscle and is involved in the regulation of the contractile function, muscle mitochondrial biogenesis, and oxidative capacity. PERM1 can be induced by exercise and is downregulated in states of physical inactivity and diseases.

#### 4.1.1. Cellular Localization

PERM1 is predominant in the cytosol and is located in the outer mitochondrial membrane of muscle cells [25]. Additionally, like PGC-1α, PERM1 is more abundant in type IIa muscle fibers, mostly in the subsarcolemmal regions [26]. PERM1 interacts with multiple proteins in the mitochondrial contact site and cristae organizing system/mitochondrial intermembrane space bridging (MICOS–MIB) complex in mouse muscles, and its ablation disrupts these interactions, reducing MICOS/MIB enrichment in subsarcolemmal mitochondria [26]. Furthermore, in mouse skeletal muscle, PERM1 binds to lamin A/C—a protein involved in the control of nuclear structures, genome integrity, tissue-specific 3D genome organization, and tissue-specific gene regulation—suggesting a potential role for PERM1 in nuclear architecture and genome organization within muscle cells [29].

#### 4.1.2. Signaling/Gene Induction

An overview of the PERM1 signaling pathway in skeletal muscle is provided in Figure 3. The pioneering study by Cho et al. (2013) characterized PERM1 in C2C12 myotubes, revealing that PERM1 acts downstream of PGC-1α and ERRs and is involved in the regulation of muscle-selective pathways of energy metabolism and in the regulation of the contractile function [12]. Overexpression of PGC-1α, PGC-1β, ERRα, ERRβ, and ERRγ in C2C12 myotubes induce *PERM1* expression, with PGC-1α and ERRα leading to a more pronounced increase in *PERM1* [12]. Interestingly, MetaMEx reveals that *PGC-1α* and *PERM1* expression is strongly correlated in human muscles [20]. Additionally, CaMKII (calcium/calmodulin-dependent protein kinase II) interacts with PERM1 forming a regulatory feed-forward loop where PERM1 promotes the activation of CaMKII, and the activation of CaMKII increases the expression and activity of *PERM1* [13]. In GLUT-4 mutated mouse skeletal muscle, PERM1 phosphorylation is increased, suggesting that PERM1 may work in collaboration with AMPK and CaMK signaling [30].

Activation of *PERM1* induces the expression of a gene set affecting a diverse range of cellular processes including energy metabolism homeostasis, mitochondrial biogenesis, lipid and glucose uptake, and contractile activity. Overexpression of PERM1 in C2C12 myotubes induces *PGC-1α* and *ERR* target genes, including *CKMT2, GLUT4*, and *TNNI3*, while PERM1 suppression increases *PDK4* mRNA, indicating its negative regulation by *PERM1* [12]. Additionally, specific *PGC-1α/ERR/PERM1*-regulated genes (see Figure 3) rely predominantly on *PERM1* for an efficient response to *PGC-1α* [12]. PERM1 overexpression has been shown to increase p-p38 MAPK protein levels in both the primary myotubes and the tibialis anterior muscle of mice, while upregulating other proteins and genes involved in the regulation of mitochondrial biogenesis and of lipid and glucose metabolism (see Figure 4) [11].

Findings from PERM1 loss-of-function studies have also contributed to the elucidation of PERM1’s downstream network. Muscle-specific PERM1 knockdown in mice has been found to reduce acute exercise-mediated increases in p-CaMKII and p-p38 MAPK protein expression, attenuate induction of exercise-responsive genes associated with oxidative capacity, and increase exercise-responsive genes associated with cellular stress (see Figure 3) [13]. Additionally, PERM1 knockdown in mice performing 4 weeks of voluntary wheel running has been shown to decrease the protein levels of all OxPhos complexes and PGC-1α, abolish training-induced increases in myoglobin, CKMT2, and CPT1B, and increase ERRα and SIRT3 [13]. Similarly, PERM1 knockdown in mice has been shown to decrease expression of proteins associated with the OxPhos function, the MIB complex, oxidative stress signaling, mitochondrial energy metabolism, and apoptosis (see Figure 3) [26].

#### 4.1.3. Mitochondrial Biogenesis

PERM1 is an indispensable modulator of maximal mitochondrial oxidative capacity and a strong enhancer of PGC-1α-mediated mitochondrial biogenesis [11,12,13]. PERM1 knockdown in mouse skeletal muscle has been found to (i) decrease SDH activity, subsarcolemmal mitochondria formation, mitochondrial content, (ii) alter the regulation of proteins involved in membrane transport and ROS breakdown, and (iii) reduce skeletal muscle strength and endurance [26]. Interestingly, PERM1-overexpressing muscles from mice have been found to exhibit a higher density of mitochondria, enhanced spare respiratory capacity, increased vascularization, and improved fatigue resistance [11]. Furthermore, after dual innervation surgery (hypoglossal nerve sutured to the masseteric nerve), rat skeletal muscle has been shown to upregulate levels of *PERM1, CKMT2*, and *CaMKII*, along with a potential shift toward a more oxidative muscle fiber profile [28].

#### 4.1.4. PERM1 and Exercise

In muscles from mice experiencing PERM1 knockdown, the exercise-induced expression of genes involved in oxidative capacity control (*PGC-1α, GADD45G, NR4A3*) has been found to be reduced [13]. Interestingly, a 5-week period of voluntary wheel training has been shown to lead to an upregulation of *PERM1* mRNA in the skeletal muscles of mice [12]. In human skeletal muscle, a single bout of endurance exercise has been shown to increase *PERM1* mRNA levels [12,23], while, endurance training, has been found to upregulate PERM1 protein content [21,22]. Additionally, the MetaMEx analyses conducted in the present study demonstrated no significant changes in *PERM1* expression following acute (HIT, resistance, aerobic) or chronic exercise (aerobic, resistance, HIT, and combined) (see Figure 4). Although the 95% confidence intervals for acute HIT and acute aerobic exercise did not overlap zero, suggesting a potential upregulation of *PERM1*, the corresponding *p*-values were not statistically significant.

#### 4.1.5. Physical Inactivity and Disease States

In states of physical inactivity and in diseases associated with muscle atrophy and mitochondrial dysfunction (e.g., amyotrophic lateral sclerosis and Duchene muscular dystrophy), where skeletal muscle *PGC-1α* and *ERRα* are reduced [39], *PERM1* expression is also reduced [12,13,21], and PERM1 protein degradation is accelerated [25]. Interestingly, the downregulation of the ESRRγ–PERM1–CKMT2 signaling axis in muscles from ovariectomized rats suggest that ESRRγ, PERM1, and CKMT2 are key players in the development of obstructive sleep apnea–hypopnea syndrome (OSAHS) and could be targeted to prevent or mitigate the condition [27]. Additionally, MetaMEx analyses revealed a significant decrease in *PERM1* mRNA levels in response to physical inactivity (see Figure 4), suggesting that exercise is an essential activator/inducer of *PERM1* expression.

### 4.2. PERM1 and Heart Tissue

We were able to find nine studies examining PERM1 in heart tissue and cardiomyocytes (summarized in Table 2). These studies demonstrate that PERM1 is strongly expressed in cardiac tissue and is involved in the maintenance of normal cardiac function, development and maturation of the heart, mitochondrial biogenesis, and oxidative capacity. PERM1 is downregulated in subjects suffering from cardiac diseases and experiencing pressure overload conditions. The signaling pathways surrounding *PERM1* regulation and a summary of genes regulated by *PERM1* in the cardiac muscle are presented in Figure 5.

#### 4.2.1. Localization/Structure

Research on mouse hearts indicates that PERM1 is highly expressed in cardiac tissue at the protein and *mRNA* levels [12,24]. PERM1 is predominantly expressed in the ventricular tissue, specifically within the sarcoplasmic reticulum and the outer mitochondrial fraction [32], mainly in the outer mitochondrial membrane [25]. Further analyses on human heart tissue have confirmed the mitochondrial location of PERM1 [25]. Furthermore, similarly to the skeletal muscle, in the heart tissue, PERM1 binds to lamin A/C, indicating an involvement of PERM1 in the control of nuclear architecture and genome organization [29].

#### 4.2.2. Regulation

In mouse heart tissue, PERM1 contains a PEST motif with consensus sequences for CK2 phosphorylation, thus PERM1 is phosphorylated by CK2 kinase, a process that regulates the stability and the turnover of PERM1 [25].

#### 4.2.3. Energy/Substrate Metabolism

PERM1 knockdown in mice hearts and isolated mitochondria has been shown to increase glycolytic and polyol intermediates and amino acids, enhance glycerolipid biosynthesis, reduce free fatty acids and monoglycerides, and reduce solute carrier membrane proteins (SLCs) [25,33]. Furthermore, PERM1 regulates several genes implicated in fatty acid and carbohydrate metabolism (see Figure 5) [34].

#### 4.2.4. Mitochondrial Biogenesis

PERM1 regulates mitochondrial biogenesis in heart tissue by modulating multiple genes indispensable for mitochondrial biogenesis, oxidative function, and energy transduction (see Figure 5) [31,32,33]. Mouse hearts and cardiomyocytes overexpressing PERM1 have been found to exhibit increased levels of PERM1, OxPhos, mitochondrial DNA content, and maximal oxygen consumption [32], whereas PERM1 knockdown has been shown to alter mitochondrial morphology, reduce OxPhos, and increase glycolysis [33]. The continuous increase in PERM1, PGC-1α/β, and ERRα/γ expression from the initial cardiac development to birth also supports the hypothesis of a role played by PERM1 in the regulation of mitochondrial function and biogenesis—PERM1 modulates this process by acting as the co-factor of ERRα and by modifying PGC-1α, ERRα, and their specific targets [31,32]. Interestingly, *PERM1* has been found to experience downregulation in the heart tissue of ERRγ knockdown mice [24].

PERM1 can increase ERR response element (ERRE) activity in cardiomyocytes, thereby inducing its own transcription and activating ERRα respective target genes [31]. This positive feedback loop promotes a stronger signal for *SMYD1*, a muscle-specific histone methyltransferase upstream *PERM1*, involved in the control of mitochondrial energetics in the heart [31,40]. Furthermore, PERM1-induced transcriptional activation via ERRE relies on the physical interaction of PERM1 with ERRα and PGC-1α, and the underexplored transcriptional regulators BAG6 and KANK2 [33].

PERM1 also interacts with PGC-1α, improves ERR-dependent transcriptional activity, and enables more recruitment of PGC-1α to target gene promoters [32]. Further, PERM1 contributes to the maintenance of PGC-1α expression through a combination of repressing PGC-1α expression at the transcription level and stabilizing the PGC-1α protein via post-transcriptional modifications (PTM) [31]. Additionally, PERM1 knockdown in cardiomyocytes impairs PPARα-mediated transcription, while PGC-1α knockdown inhibits PERM1-induced transcriptional activation via PPRE, indicating that PERM1 depends on PGC-1α to promote PPARα transcriptional activity [34].

#### 4.2.5. Diseases and Pressure Overload

Cardiac tissue from failing hearts in both mice and humans exhibits dysfunction associated with reduced levels of PERM1 *mRNA* and protein [31,32]. PERM1 deletion may contribute to the heart failure by altering mitochondrial morphology, decreasing OxPhos protein levels, and increasing glycolysis [33]. Thus, PERM1 is emerging as a promising target for preventing cardiac mitochondrial damage caused by ischemia–reperfusion injury by regulating OxPhos proteins [32].

In rodent models under pressure overload, the myocytes and the hearts exhibit downregulated levels of *PERM1, ERRα*, and *PGC-1α*, along with a reduced transcriptional regulation of ERR target genes [31,33]. Maintaining PERM1 expression prevents energetic defects by restoring the expression of mitochondrial biogenesis genes, including *ERRα* and some *OxPhos* components [31]. Therefore, since *PERM1* downregulation may be a primary event in the metabolic adaptation to hemodynamic stress, preserving *PERM1* expression could minimize the impact of pressure overload on the ERRα/PGC-1α axis [33]. Furthermore, a study on rodent cardiomyocytes found that PERM1 binds to Mic60—a protein part of the MICOS complex—and protects the heart against pressure overload by restoring oxidative metabolism [35].

### 4.3. PERM1 and Other Tissues

The role of PERM1 in other tissues has received far less attention than the role of PERM1 in skeletal/heart muscle. Our review revealed only a handful of studies reporting the effects of PERM1 in adipose tissue, tumors, HEK293 cells, blood cells, and brain tissue. We found seven studies examining PERM1 in these tissues (summarized in Table 3).

PERM1 *mRNA* and protein levels are reduced in mouse brown adipose tissue [12,24], but its *mRNA* levels increase in in vitro differentiated brown adipocytes following PGC-1α overexpression [24]. PERM1 is crucial for brite/beige adipocyte formation, both in vitro and in vivo, by regulating OxPhos components, while ERRγ regulates UCP1 levels [24]. Additionally, *ERRγ* and *PERM1* levels rise in the inguinal white adipose tissue of mice exposed to cold temperatures [24].

Studies on desmoid-type fibromatosis (DTF) tumors, including S45F and T41A, have identified *PERM1* as a gene associated with differentially methylated regions between these tumor subtypes [36]. Moreover, in HEK-293 cells, PERM1 anchors into the outer mitochondrial membrane via its C-terminal domain and binds to the CaMKII protein [13,26], suggesting that CaMKII and PERM1 may be part of the same multi-protein complex and interact with each other via the outer mitochondrial membrane. PERM1 has been used as a marker of mitochondrial biogenesis in human peripheral white blood cells from acute intermittent porphyria patients, with lower serum PERM1 levels and mtDNA content being observed in patients compared to healthy controls [37]. Finally, a study on rodent neurons after an ischemic stroke found that miR-214-3p regulates *PERM1* at the transcriptional level, affecting mitochondrial function and inflammatory responses, highlighting the role of the miR-214-3p/PERM1 axis in mitochondrial damage and inflammation during neuronal injury [38].

## 5. Limitations and Future Directions

Our risk of bias assessment revealed rates of high/unclear risk of bias of over 80% for selection, performance, detection, attrition, and reporting bias (See Appendix A). Importantly, inadequate protection against experimental bias can contribute to inflated and confirmatory treatment effects [41,42] and is fueling the growing reproducibility crisis [43,44]. These results align with our recent observations of widespread risk of bias in exercise science [19] and highlight a need for improved experimental rigor and reporting practices in future PERM1 studies. Future studies on PERM1—and across basic/exercise science—should implement and report randomization and blinding procedures, proactive attrition management, and complete pre-registration.

The bulk of the available evidence supporting the role of PERM1 in the regulation of skeletal muscle mitochondrial content comes from animal and cell models (e.g., over- and underexpression experiments). Where human data are available, they present only superficial information on PERM1 function. Specifically, human studies have observed changes in *PERM1* mRNA expression after a single bout of exercise [12,23], altered protein expression after training [22] and in endurance-trained individuals [21], and altered protein expression in diseased muscle [12,13]. Although these data are mostly consistent with a positive relationship between PERM1 and mitochondrial content in human muscle, data that comprehensively examine the conservation of the PERM1 pathway described in mice—including regulatory signaling and downstream gene set—are not currently available. Thus, there is a need for studies that characterize the PERM1 pathway (see Figure 3) and its relationship with exercise-mediated increases in mitochondrial content in human skeletal muscle.

PERM1 shows promise as a therapeutic target for treatment/prevention of heart disease [32,33,35]; however, significant barriers remain before preliminary findings can be translated into clinical practice. Importantly, most existing research connecting PERM1 with heart failure comes from animal/cell models, with only two human studies demonstrating decreased PERM1 protein levels in cardiac tissue from patients with dilated cardiomyopathy and heart failure [31,32]. Thus, barriers to the translation of PERM1 research into clinical practice include the need for more human heart studies to clarify its role in mitochondrial function during heart failure, the lack of clinical trials investigating its therapeutic potential, and challenges associated with the development of targeted methods to modulate PERM1 expression without causing off-target effects. Although there is currently no direct evidence supporting the efficacy of therapeutic modulation of PERM1 in clinical settings, the therapeutic potential of PERM1 represents an intriguing and potentially exciting avenue for future research.

An additional limitation specific to the human PERM1 literature is the lack of inclusion of female participants or consideration of sex differences. Of the 10 human studies included in this review, four failed to report the sex of their participants, three studied only males, and three included males and females. No studies were designed to investigate sex-specific differences in PERM1 expression, the regulation of the PERM1 pathway, or the relationship between PERM1 and mitochondrial content. Studies designed to assess sex differences in PERM1 expression/regulation of PERM1 in human muscle are needed.

## 6. Conclusions

The current review provides a summary of the current state of knowledge surrounding PERM1 function. In skeletal muscle, PERM1 contributes to the regulation of mitochondrial biogenesis through a regulatory loop that includes PGC-1α. In human muscle, PERM1 is generally upregulated following exercise and downregulated in diseased patients and following inactivity. Whether there are specific conditions where PERM1 is more or less important than other transcriptional regulators remains unknown. However, evidence from culture myotubes suggests that Perm1 is required for PGC-1α-induced mitochondrial biogenesis [12,13] and a feed-forward regulatory loop including both PGC-α and PERM1 exists. Thus, it appears that PERM1 and PGC-1a likely play complementary roles in the regulation of mitochondrial biogenesis. In cardiac tissue, PERM1 is essential for both development and maturation of the heart, maintenance of normal cardiac function, and mitochondrial biogenesis. PERM1 also provides protection against cardiac and mitochondrial damage following ischemia–reperfusion injury and pressure overload. There is a need for rigorously performed studies focused on PERM1 function in human tissue and for further exploration of the potential therapeutic potential of PERM1 in humans.

## Figures and Tables

**Figure 1 genes-15-01305-f001:**
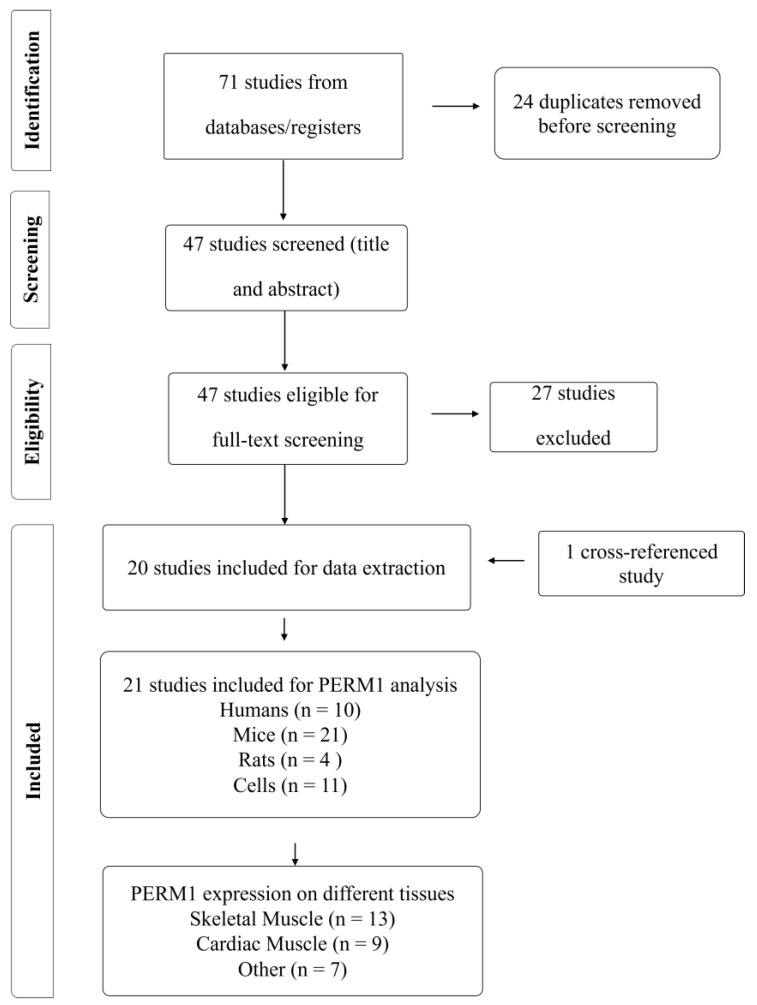
Flow chart of study selection process including revised eligibility criteria.

**Figure 2 genes-15-01305-f002:**
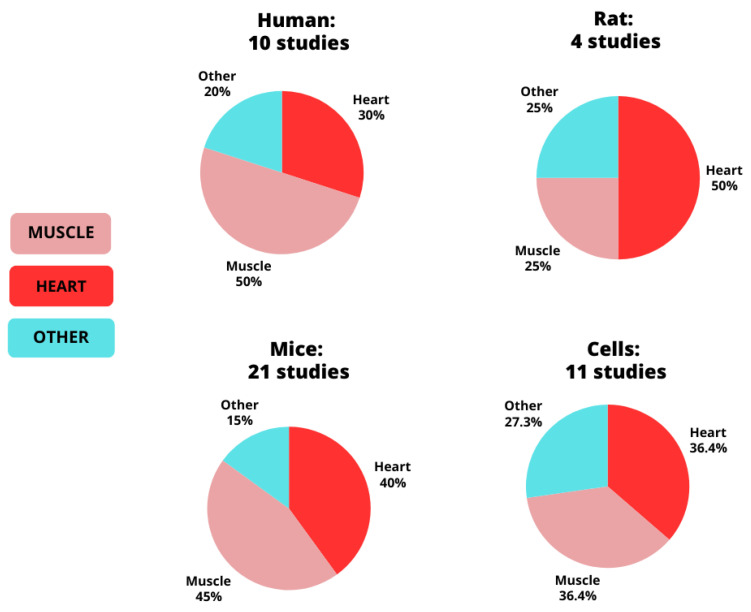
Distribution of PERM1 studies by species and tissue type.

**Figure 3 genes-15-01305-f003:**
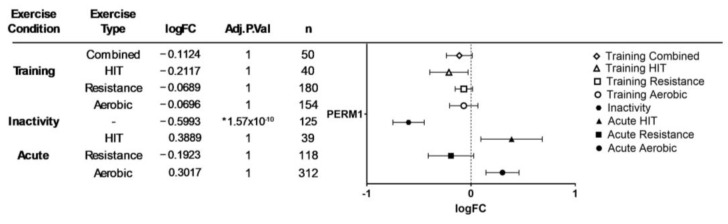
Left: meta−analyzed expression data of *PERM1* mRNA levels in human skeletal muscle after acute and chronic (training) aerobic exercise, resistance exercise, HIT exercise, combined training, and physical inactivity. Right: forest plot. Log fold change and 95% confidence intervals of *PERM1* levels after acute and chronic (training) aerobic exercise, resistance exercise, HIT exercise, combined training, and physical inactivity. Symbols denote significant (* *p* ≤ 0.05) differences in *mRNA* expression. See Appendix A for additional information.

**Figure 4 genes-15-01305-f004:**
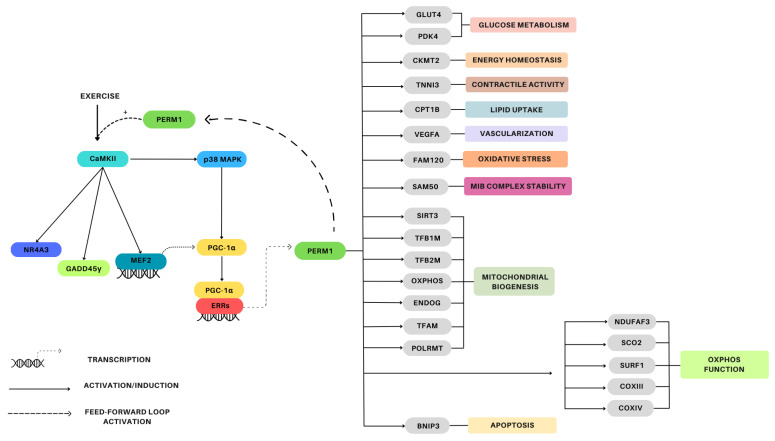
Signaling pathways surrounding PERM1 regulation and a summary of genes regulated by PERM1 in skeletal muscle.

**Figure 5 genes-15-01305-f005:**
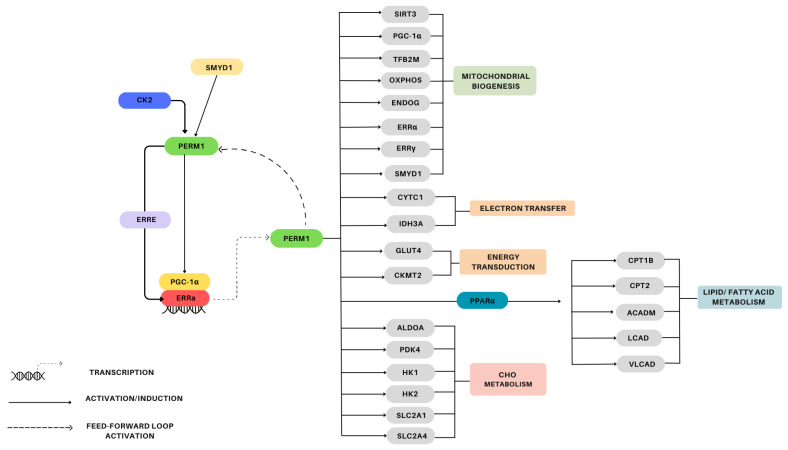
Signaling pathways surrounding PERM1 regulation and a summary of genes regulated by PERM1 in the cardiac muscle.

**Table 1 genes-15-01305-t001:** Studies examining PERM1 in skeletal muscle.

Author	Model/Intervention	Key Findings
Cho et al., 2013 [12]	C2C12 cells	Increases in PERM1 following PGC-1α/β overexpression are dependent on ERRα; PGC-1α and ERRα bind to ERRE2 on the PERM1 promotor. PERM1 protein is primarily restricted to nuclear and cytosolic fractions. PERM1 overexpression upregulates some (*CKMT2*, *TNNI3*, and *GLUT4*), but not all, PGC-1 and ERR targets and does not alter expression of PGC-1 and ERR family members, suggesting that PERM1 does not act by altering general PGC-1 or ERR levels and activity. PERM1 knockdown does not affect basal mtDNA but does reduce basal OxPhos (*mRNA* and protein) expression, and blunt PGC-1α induced- increases in mtDNA and OxPhos (*mRNA* and protein). PERM1 knockdown downregulates *CPT1b* mRNA after PGC-1α induction.
	Mice	*PERM1* mRNA and PERM1 protein are highly expressed in skeletal muscle, heart, and, to a lesser degree, brown adipose tissue. *PERM1* increases in quadriceps muscle following 5 weeks of voluntary running.
	Human	Skeletal muscle *PERM1* mRNA increases following one bout of endurance exercise (60 min @ 70%VO_2max_) and is reduced in the muscles of patients with ALS.
[21]	Human	An increased abundance of PERM1 and CKMT2 proteins is found in the skeletal muscle, at rest, of endurance-trained subjects compared with untrained/sedentary individuals.
[11]	Muscle specific PERM1 gain-of-function in TA of mice	PERM1 does not impact on muscle morphology, fiber size, or fiber type. PERM1 increases mtDNA, OxPhos protein/activity, CS and SDH activity, mitochondrial content, maximal mitochondrial respiration, capillary density, and vascularization (VEGFA protein expression). PERM1 increases PGC-1α, ERRα, SIRT3 *mRNA*, protein levels, the expression of *TFAM*, *TFB2M*, *ENDOG*, *GLUT4*, and *CPT1B* genes, and MYOGLOBIN protein expression. PERM1 increases p-p38 MAPK during active hours and enhances fatigue resistance in skeletal muscle.
	Primary Myotubes	PERM1 increases p-p38 MAPK in primary myotubes.
[22]	Human	Endurance exercise training leads to PERM1 protein upregulation in the skeletal muscle of patients with type 2 diabetes mellitus. PERM1 is homogenously distributed in type I and type II muscle fibers of patients with type 2 diabetes mellitus prior to exercise training.
[23]	Human	Upregulation of skeletal muscle *PERM1* mRNA and *PGC-1a* mRNA occurs following an acute bout of exercise (three all-out cycle sprints of 30 s duration interspersed with 20 min recovery).
[13]	Mice	PERM1 co-precipitates with CaMKII in mouse TA muscles. PERM1 activation of *MEF2* promotor is CaMKII-dependent.
	PERM1 knockdown in mice	In the skeletal muscle of mice performing acute exercise, PERM1 knockdown leads to: Reduced exercise-mediated increases in p-CaMKII and p-p38 MAPK protein expression. Reduced induction of *PGC-1α*, *NR4A3*, and *GADD45G* and increased induction of *ATF3* and *MT2* gene expression.
		In the skeletal muscle of mice performing 4 weeks of voluntary wheel running, PERM1 knockdown leads to: Decreased training induces an increase in mtDNA content. Decreased protein levels of all OxPhos complexes in both sedentary (SED) and trained (TR) mice. Reduced activities of complex IV, SDH, and citrate synthase in both SED and TR mice. Abolished training induces increases in myoglobin, CKMT2, and CPT1B proteins, decreases in the protein levels of PGC-1α, and increases in ERRα and SIRT3 proteins in both SED and TR muscles.
	Mice—Diet induced Obesity	Skeletal muscle *PERM1* mRNA and protein levels decreases in diet-induced obese mice.
	Duchenne muscular dystrophy (human)	Skeletal muscle *PERM1* mRNA levels decrease in muscular dystrophy patients and in the tibialis anterior muscle of mouse models of Duchenne muscular dystrophy.
[24]	Mice	*PERM1* mRNA and protein levels are highly expressed in skeletal muscle.
[25]	C2C12 cells	PERM1 is located in the outer mitochondrial membrane.
	Mice	Enhanced protein synthesis of PERM1 occurs in the skeletal muscle of denervation-induced atrophy mouse.
[26]	PERM1 knockdown in mice	Skeletal muscle PERM1 protein is upregulated during the night when mice are more physically active. PERM1 is abundant in the subsarcolemmal regions of type IIa muscle fibers and this abundance correlates with increased PGC-1α. PERM1 protein expression is more abundant in SSM than in IFM in wild-type TA muscles. PERM1 interacts with ankyrin B (ANKB), an adaptor protein that interacts with the cytoskeleton and connects transmembrane proteins with the sarcolemma. Deletion of PERM1 leads to: Reduced muscle strength and endurance without fiber-type switches or changes in fiber diameter in skeletal muscle. Decreased skeletal muscle SDH activity, mtDNA copy number, and mitochondria:capillary ratio. Reduced formation of subsarcolemmal (SS) mitochondrial and decreased expression of the members of the MICOS–MIB complex. Intermyofibrillar mitochondria remain unaffected. Downregulated SCO2, NDUFAF3, SURF1, and COX3 (proteins associated with OxPhos function and OxPhos assembly). Decreased mitochondrial levels of SAM50, FAM120, and BNIP3 proteins. Decreased skeletal muscle POLRMT protein expression during the night. No changes in mitochondrial fusion/fission, autophagy, and mitophagy markers.
	C2C12 cells	PERM1 interacts with the MICOS–MIB complex.
[27]	Ovariectomized Rats	A rat model of obstructive sleep apnea–hypopnea syndrome (ovary removal) shows lower blood estrogen levels and reduced protein expression of PERM1, ESRRG, CKMT2 in the genioglossus muscle.
[28]	Rat	Dual innervation technique surgery leads to the upregulation of *PERM1, CKMT2*, and *CAKMII* in masseter muscle samples from rats.
[29]	Mice	PERM1 strongly binds to lamin A protein.
[30]	Mice	Glucose restriction increases the phosphorylation levels of PERM1 in the skeletal muscle.

**Table 2 genes-15-01305-t002:** Studies examining PERM1 in the cardiac muscle.

Author	Model/Intervention	Key Findings
[12]	Mice	Increased expression of *PERM1* mRNA in heart tissue.
Oka et al., 2020 [31]	PERM1 gain-of-function in cardiomyocytes	PERM1 overexpression represses the *PGC-1α* gene (decreased PGC-1α promoter activity and *mRNA* levels) and increases PGC-1α protein levels. PERM1 has the capacity to increase ERR response element (ERRE) activity.
	PERM1 knockdown in cardiomyocytes	Decreased PGC-1α protein, followed by decreased levels of *PGC-1α* mRNA.
	SMYD1 knockdown in neonatal rat ventricular myocytes	*PERM1* mRNA downregulation, suggesting that *PERM1* is a target gene of *SMYD1*.
	PERM1 knockdown in neonatal rat ventricular myocytes	Downregulation of ERRα and PGC-1α proteins. Downregulation of the tricarboxylic acid (TCA) cycle enzymes (Sdhb, Idh2), the subunits of protein complexes comprising the electron transport chain (ETC) (Ndufb8, Ndufs8, Cox5a, Cox8a, Cox6a1, Coq2, Coq10b), the transporter and enzyme of fatty acid β-oxidation (FAO) (Cpt1b, Acsl1), and the glycolytic enzymes (Eno3, Pdgdh). Upregulation of genes involved in the biosynthesis of amino acids (Psph, Pycr2), sphingolipid metabolism (Gnai2, Pik3ca, Asah2, Cers5, Kdsr, Sgms1), and the phosphatidylinositol signaling system (Inpp5k, Ipmk, Mtmr11, Pik3ca). PERM1 knockdown in cardiomyocytes leads to the downregulation of genes encoding the ETC subunits (*Ndufv1, Ndufs1*), the TCA cycle (*Cs*, *Sdh3b*, *Idh3b*), and the enzymes and transporters involved in FAO (*Cpt1b*, *MCAD*, *Acox1*) and glucose utilization (*Gult4, Pdhb*). Downregulation of *ERRα*, *PGC-1α*, *PPARα*, and *NRF1*. Reduced basal respiration rate and ATP production, suggesting that PERM1 is required to maintain mitochondrial energetics in cardiomyocytes.
	PERM1 gain-of-function in neonatal rat ventricular myocytes	Upregulation of *ERRα, Ndufv1, Ndufs8* (Complex I), ATP5a (Complex VI), *CPT1b*, and *CPT2* (fatty acid β-oxidation).
	Neonatal rat ventricular myocytes treated with phenylephrine (hypertrophic stress/pressure overload model)	Under pressure overload, neonatal ventricular myocytes demonstrate lower *PERM1, ERRα*, and *PGC-1α*. Under pressure overload, when *PERM1* expression is maintained in neonatal ventricular myocytes, the expression of *ERRα* and some *OxPhos* genes is restored, thus preventing energetic defects.
	Mouse failing hearts	Decreased expression of PERM1 protein in cardiac tissue from failing hearts.
	Human failing hearts	Cardiac tissue from failing hearts demonstrates reduced expression of PERM1 protein.
[24]	Mice	PERM1 *mRNA* and protein levels are highly expressed in heart tissue.
		*PERM1* is downregulated in the heart tissue of ERRγ knockdown mice.
[32]	Cardiomyocytes	PERM1 gain-of-function increases mitochondrial DNA content, OxPhos protein levels, and maximal oxygen consumption. PERM1 can improve both mitochondrial biogenesis and oxidative metabolism by modifying PGC-1α and ERR levels and, consequently, its specific targets. PERM1 regulates PGC-1/ERR targets (*PGC-1α, ERRα, ERRγ, TFB2M, SIRT3*, and *ENDOG*) and genes essential for electron transfer (*CYTC1 and IDH3A*) and energy transduction (*GLUT*4 and *CKMT2*). PERM1 has the capacity to prevent the cellular damage generated by hypoxia reoxygenation (ischemia–reperfusion injury model) through the regulation of OxPhos proteins.
	Mice	PERM1 protein is more predominant in the ventricular tissue, specifically within the sarcoplasmic reticulum and mitochondrial fraction. Developing hearts from mice demonstrate upregulated PERM1 and OxPhos proteins and increased mitochondrial DNA copy number. PERM1 interacts with PGC-1α, improves ERR-dependent transcriptional activity, and enables more recruitment of PGC-1α to target gene promoters. *PERM1, PGC-1s* (α and β), and *ERRs* (α and γ) increase continuously from cardiac development to birth. Decreased *PERM1* mRNA and protein detected in heart failure model are associated with cardiac dysfunction.
	Human	Ventricular tissue from patients with dilated cardiomyopathy demonstrate reduced expression of PERM1 protein.
[25]	Isolated mitochondria from mouse hearts	PERM1 is co-located with the mitochondrial marker TOM20, indicating that PERM1 is located in the outer mitochondrial membrane.
	Hearts of mice	PERM1 protein levels increase during early heart development after birth. PERM1 is phosphorylated by CK2 kinase.
	PERM1 knockdown mice	Reduced protein expression of LPIN1—modulator of lipid homeostasis—in the mitochondrial fraction of the heart tissue. Heart and isolated mitochondria demonstrate reduced levels of solute carrier membrane proteins (SLCs) and increased levels of amino acids.
	Human	PERM1 is more predominant in the outer mitochondrial membrane of the human heart tissue.
[33]	Mice	In PERM1-deficient hearts, mitochondrial cristae demonstrate reduced density and increased disorganization, reduced OxPhos protein levels, and increased glycolysis. In mouse hearts under pressure overload, PERM1-induced transcriptional regulation of ERR target genes is reduced. Hearts from PERM1 knockdown mice show increased accumulation of glycolytic and polyol intermediates, enhanced glycerolipid biosynthesis, and reduced levels of free fatty acids and monoglycerides.
	Cardiomyocytes	PERM1 physically interacts with ERRα, ANKRD1, BAG6, KANK2, and TIF1β.
[34]	Cardiomyocytes	PERM1 knockdown in cardiomyocytes impairs PPARα-mediated transcription, while PGC-1α knockdown inhibits PERM1-induced transcriptional activation via PPRE.
	Mice	PERM1 knockdown mouse hearts experience downregulation of several genes implicated in fatty acid metabolism (*CPT1β, CPT2, ACADM, LCAD*, and VLCAD) and carbohydrate metabolism (*SLC2A1, SLC2A4, HK1, HK2, PDK4*, and ALDOA). The transcription of genes implicated in fatty acid metabolism in mouse hearts is mediated by PERM1 interactions with PPARα and PGC-1α.
[29]	Mice	PERM1 binds to lamin A protein.
[35]	Mice	Pressure overload in hearts overexpressing PERM1 causes upregulated levels of MICOS and OxPhos proteins and increased mitochondrial DNA copy number. PERM1 binds to the Mic60 protein in heart lysates. In cardiomyocytes: Decreased Mic60 expression reduces MICOS and OxPhos protein levels. PERM1 overexpression increases Mic60 and Mic10 protein levels and maintains the levels of MICOS proteins, when Mic60 shRNA is induced. Mic60 loss-of-function decreases the maximal oxygen consumption rate even when PERM1 is overexpressed.

**Table 3 genes-15-01305-t003:** Studies examining PERM1 in other tissues.

Author	Model/Intervention	Key Findings
[12]	Mice	Reduced *PERM1* mRNA and protein levels in brown adipose tissue.
Cho et al., 2019 [13]	HEK293 cells	PERM1 co-precipitates with CaMKII.
[24]	Adipose	Low *PERM1* expression in in vitro differentiated adipocytes, followed by upregulation of *PERM1* upon adenoviral overexpression of PGC-1α.
	Immortalized brown preadipocytes	PERM1 (at the *gene* and protein level) expression upon doxycycline treatment leads to increased OxPhos protein expression.
	Mice adipose tissue	*PERM1* and *ERRγ* are upregulated in the inguinal white adipose tissue mature adipocyte fraction of mice after 24 h of exposure to cold temperatures. *PERM1* expression positively correlates with browning markers. PERM1 protein is expressed after cold stimulation in inguinal white adipose tissue, indicating that the changes at the transcript level also result in increased protein abundance. PERM1 *mRNA* and protein are highly expressed in intrascapular brown adipose tissue. In vivo overexpression of PERM1 in mice, combined with exposure to cold temperatures, increases the levels of selective OxPhos components (Complex 1, 3, and 5).
[36]	Human desmoid-type fibromatosis (DTF) tumors	*PERM1* is associated with the differentially methylated regions between S45F and T41A tumors.
[26]	HEK-293T cells	PERM1 anchors into the outer mitochondrial membrane via its C-terminal domain.
[37]	Human white blood cells	Peripheral white blood cells from patients with acute intermittent porphyria exhibit lower levels of PERM1 and mtDNA content compared with healthy controls.
[38]	Neurons	In neurons, following an ischemic stroke, miR-214-3p regulates *PERM1* expression at the transcriptional level, influencing mitochondrial function and inflammatory responses. The miR-214-3p/PERM1 axis is a critical pathway involved in mitochondrial damage and in the inflammatory response during neuronal injury caused by an ischemic stroke.

## Data Availability

The original contributions presented in the study are included in the article/Appendix A. Further inquiries can be directed to the corresponding author.

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
