# Peer review of "PERM1—An Emerging Transcriptional Regulator of Mitochondrial Biogenesis: A Systematic Review"

_genes, 2024, doi:10.3390/genes15101305_

Round 1

Reviewer 1 Report

Comments and Suggestions for Authors

The article entitled "PERM1 - An Emerging Transcriptional Regulator of Mitochondrial Biogenesis: A Systematic Review” aims to investigate the role of PERM1 in different organisms, tissues and cellular functions, with particular emphasis on its involvement in the regulation of skeletal muscle mitochondrial biogenesis.

The review discusses the role of PERM1 in mitochondrial biogenesis, but to what extent does this review provide new insights beyond previous research? Are there specific findings that significantly advance our understanding?

The review mentions the use of Covidence software for data extraction. Can you explain how inter-rater reliability was ensured during the screening and extraction process?

"Was a meta-analysis considered and if not, can you justify why a narrative synthesis was preferred despite the quantitative nature of some studies?

The review focuses on mouse models and exercise-induced mitochondrial changes. How transferable are these results to human physiology, especially given the limited number of human studies?

Since the role of PERM1 in human cardiac tissue under stress is emphasized, should more emphasis have been placed on human disease models in the review?"

The Cochrane Collaboration's Risk of Bias Tool was used, but the document does not address potential bias in detail. Were significant risks of bias identified and how were they addressed in the analysis?

The review suggests that PERM1 may be a therapeutic target for heart failure. What are the main barriers to translating these findings into clinical practice, and is there evidence from the studies reviewed that therapeutic targeting of PERM1 is possible?

Given that PGC-1α is traditionally considered a master regulator of mitochondrial biogenesis, how does PERM1 fit into this framework? Are there specific conditions in which PERM1 plays a more important role than PGC-1α, and were these adequately explored in the review?

Does the review take into account possible sex-specific differences in PERM1 expression and function, particularly in human studies? If not, should this be considered to improve the generalizability of the results?

Author Response

Comment 1. The review discusses the role of PERM1 in mitochondrial biogenesis, but to what extent does this review provide new insights beyond previous research? Are there specific findings that significantly advance our understanding?

Response 1: The current manuscript primary adds to our understanding of PERM1 function through a comprehensive synthesis of previous work. However, as highlighted in the reviewer’s comments below, our review also highlights important gaps in the literature and exciting future directions.  To highlight both the importance of the current synthesis, and the gaps/future directions our review has revealed, we have revised our conclusion and added a “limitations and future directions” section to the end of our manuscript.

Please see our revised manuscript (Pages 01-21) and specific details of our responses to the reviewer’s questions below.

Comment 2. The review mentions the use of Covidence software for data extraction. Can you explain how inter-rater reliability was ensured during the screening and extraction process?

Response 2: To improve the transparency of our methodology we have included the following details in “2.2. Search strategy”, page 2 and 3, lines 89-98:

“Two independent reviewers evaluated each article during both the screening and data extraction phases using Covidence software. In an attempt to ensure consistency in the review process, all reviewers were familiarized to our inclusion criteria prior to the start of screening/extraction, and discussions were had throughout the review process as questions/conflicts arose. During the data extraction phase, reviewers utilized a data extraction template generated on Covidence.

Any conflicts that arose during the screening and extraction process were resolved through discussion between the two reviewers who initially screened/extracted data from an article.  If consensus could not be reached through discussion, a third reviewer was consulted to resolve the conflict.”

Comment 3. Was a meta-analysis considered and if not, can you justify why a narrative synthesis was preferred despite the quantitative nature of some studies?

Response 3. A meta-analysis was considered during the planning stages of this review. However, we ultimately decided that a meta-analytical approach was not warranted given the high heterogeneity among the included studies regarding study design, differences in species and type of tissues, methodologies, outcome measures, and interventions.

To prevent a misleading and/or underpowered meta-analysis we chose a narrative synthesis as a more appropriate method to comprehensively summarize and interpret findings across studies. This allowed for a broader, context-rich exploration of the data, highlighting trends and patterns while considering the diversity of the studies included.

To clarify this point to the reader, we included the following details in “2.5. Data Extraction and Data Synthesis”, page 3, lines 122-127:

“A meta-analysis was considered during the planning stages of this review. However, we ultimately decided that a meta-analytical approach was not warranted given the high heterogeneity in study design/methodology, species/tissue type, and outcome measures across available studies. Thus, to prevent a misleading and/or underpowered meta-analysis a narrative synthesis was selected as a more appropriate method to comprehensively summarize findings across studies.”

Comment 4. The review focuses on mouse models and exercise-induced mitochondrial changes. How transferable are these results to human physiology, especially given the limited number of human studies?

Response 4: This is an excellent point, and one we are particularly sensitive to given both our lab’s focus on human muscle physiology and repeated past demonstrations of a lack of conservation of molecular function between mice and human muscle. We apologize for not highlighting this issue in the initial draft of our manuscript.

In an attempt to better highlight the importance of future work in human tissue we have included a section on this topic in our new “limitations and future directions” section.  Pages 18-19, Lines 442-454, now read:

“The bulk of the available evidence supporting a role for PERM1 in the regulation of skeletal muscle mitochondrial content comes from animal and cell models (eg. over- and under-expression experiments). Where human data is available it presents only superficial information on PERM1 function. Specifically, human studies have observed changes in PERM1 mRNA expression after a single bout of exercise (Cho et al. 2013; Rundqvist et al. 2019), altered protein expression after training (Brinkman et al. 2019) and in endurance trained individual (Schild et al. 2015), and altered protein expression in diseased muscle (Cho et al. 2013; Cho et al. 2019). Although these data are mostly consistent with a positive relationship between PERM1 and mitochondrial content in human muscle, data that comprehensively examines the conservation of the PERM1 pathway described in mice – including regulatory signalling and downstream gene set - is not currently available. Thus, there is a need for future studies characterizing the PERM1 pathway (see Figure 3) and its relationship to exercise-mediated increases in mitochondrial content in human skeletal muscle.”

Comment 5. Since the role of PERM1 in human cardiac tissue under stress is emphasized, should more emphasis have been placed on human disease models in the review?

Response 5: Because the aim of our review was to “comprehensively examine the role of PERM1 in various organisms, tissues, and cellular functions with a focus on its putative role in the regulation of skeletal muscle mitochondrial biogenesis” we don’t believe a greater emphasis on human disease models is appropriate. 

However, we agree that we paid inadequate attention to the potential importance of PERM1 as a therapeutic target in human cardiac tissue.  Please see our response to the reviewer’s question #7 below for details on our attempt to address the role of PERM1 in cardiac disease more thoroughly.

Comment 6. The Cochrane Collaboration's Risk of Bias Tool was used, but the document does not address potential bias in detail. Were significant risks of bias identified and how were they addressed in the analysis?

Response 6: Based on the reviewers comment we have included more detail on our risk of bias assessment in the text of our manuscript.  We added the following paragraph to our results section on Pages 4 and 5, lines 196-203:

“3.1. Risk of Bias

Full details for our risk of bias analysis are presented in Supplemental Document, S3. We observed high rates of high/unclear risk of bias for Selection bias (sequence generation, 95% high/unclear; allocation concealment, 86% high/unclear), Performance bias (95% high/unclear), Detection bias (95% high/unclear), Attrition bias (86% high/unclear) and Reporting bias (100% unclear). None of the studies were judged to have low risk for all sources of bias. Thirteen out of 21 studies were judged to have unclear risk for all sources of bias. Only 1 study reported performing a sample size calculation.”

We have also added the following text to the new “limitation and future directions” section on Page 18, lines 433-441:  

Our risk of bias assessment revealed rates of high/unclear risk of bias of over 80% for selection, performance, detection, attrition, and reporting bias (See Supplemental Document, S3). Importantly, inadequate protection against experimental bias can contribute to inflated and confirmatory treatment effects (Schulz et al. 1995; Moher et al. 1998) and is fuelling the growing reproducibility crisis (Munafò et al. 2017; Baker, 2016). These results align with our recent observations of widespread risk of bias in exercise science (Preobrazenski et al. 2024) and highlight a need for improved experimental rigour and reporting practices in future PERM1 studies.  Future studies on PERM1 – and across basic/exercise science – should implement and report randomization and blinding procedures, proactive attrition management, and complete pre-registration.

Comment 7. The review suggests that PERM1 may be a therapeutic target for heart failure. What are the main barriers to translating these findings into clinical practice, and is there evidence from the studies reviewed that therapeutic targeting of PERM1 is possible?

Response 7: In response to this comment, we have made the following changes to our manuscript: 

First, we have removed mention of the potential therapeutic targeting of PERM1 from our abstract – we agree with the reviewer that many barriers remain before there is a clinical impact of PERM1 activation in vivo.  Although we have removed this idea from our abstract, we do believe that preliminary results do position PERM1 as an exciting target for future research.

Second, we have expanded discussion of the potential for PERM1 to emerge as a therapeutic target in cardiac disease in our “limitations and future direction” section.  Page 18, lines 455-467 now read:

“PERM1 shows promise as a therapeutic target for treatment/prevention of heart disease (Cho et al. 2021; Oka et al.  2022; Tachibana et al. 2023), however, significant barriers remain before preliminary findings can be translated into clinical practice. Importantly, most existing research connecting PERM1 with heart failure comes from animal/cell models with only two human studies demonstrating decreased PERM1 protein in cardiac tissue from patients with dilated cardiomyopathy and heart failure (Oka et al. 2020, Cho et al. 2021). Thus, barriers to translating PERM1 research into clinical practice include the need for more human heart studies to clarify its role in mitochondrial function during heart failure, the lack of clinical trials investigating its therapeutic potential, and challenges in developing targeted methods to modulate PERM1 expression without causing off-target effects. Although there is currently no direct evidence supporting the efficacy of therapeutic modulation of PERM1 in clinical settings, the therapeutic potential of PERM1 represents an intriguing and potentially exciting avenue for future research.”

Finally, we have added the following text to the new “conclusions” section.  Page 19, lines 485-488 now read:

“In cardiac tissue, PERM1 is essential for development and maturation of the heart, maintenance of normal cardiac function, and mitochondrial biogenesis. PERM1 also provides protection against cardiac and mitochondrial damage following ischemia-reperfusion injury and pressure overload.”

Comment 8. Given that PGC-1α is traditionally considered a master regulator of mitochondrial biogenesis, how does PERM1 fit into this framework? Are there specific conditions in which PERM1 plays a more important role than PGC-1α, and were these adequately explored in the review?

Response 8: We believe that PGC-1 α contributes to – but is not the sole/master regulator of – mitochondrial biogenesis. Rather, our view is that a network of transcriptional regulators determines the mitochondrial response to exercise/energetic stress (please see Islam et al. 2018 [PMID 29126696]; and Islam et al. 2020 [PMID 31158323] for our reviews on this topic).

The specific conditions where PERM1 is more or less important are difficult to determine. However, evidence from culture myotubes suggests Perm1 is required for PGC-1α induced mitochondrial biogenesis and maximal oxidative capacity (Cho et al. 2013; Cho et al. 2019). Based on these findings and the feed-forward regulatory loop where PGC-1a and ERRs induce PERM1, and PERM1 promotes PGC-1a and ERRa expression (Cho et al. 2015), we suggest that PERM1 does not play a more important role than PGC-1a but rather PERM1 and PGC-1a play complimentary roles in the regulation of mitochondrial biogenesis. Furthermore, we believe that PERM1 is as essential as other common transcriptional regulators within the regulatory network controlling mitochondrial biogenesis (i.e. PGC-1α/β, NRF1/2, p53, TFAM, ERR α/γ, PPARβ, etc).

To address this point, we included the following details in our revised conclusion (Page 19, lines 480-485):

“Whether there are specific conditions where PERM1 is more or less important than other transcriptional regulators is unknown. However, evidence from culture myotubes suggests Perm1 is required for PGC-1α induced mitochondrial biogenesis (Cho et al. 2013; Cho et al. 2019) and a feed-forward regulatory loop including both PGC-1a and PERM1 exists. Thus, it appears that PERM1 and PGC-1a likely play complimentary roles in the regulation of mitochondrial biogenesis.”

Comment 9. Does the review take into account possible sex-specific differences in PERM1 expression and function, particularly in human studies? If not, should this be considered to improve the generalizability of the results?

Response 9: These are excellent questions raised by the reviewer.  We have added a paragraph to our “limitations and future directions” section addressing sex differences and PERM1.  Page 19, lines 468-474 now read:

“An additional limitation specific to the human PERM1 literature is the lack of inclusion of female participants or consideration of sex-differences. Of the 10 human studies included in this review, 4 failed to report the sex of their participants, 3 studied only males, and 3 included males and females. No studies were designed to investigate sex-specific differences in PERM1 expression, the regulation of the PERM1 pathway, or the relationship between PERM1 and mitochondrial content. Studies designed to assess sex-differences in PERM1 expression/regulation of PERM1 in human muscle are needed.

Reviewer 2 Report

Comments and Suggestions for Authors

Comments:

1. In Figure 1: please use "skeletal muscle" for muscle (n-13), "cardiac muscle" for heart (n=9).

2.  In Table 2: please use "cardiac muscle" for heart tissue.

3. In Table 3: for mice study which is focused on adipose tissues, please add "adipose" in model.

4. In Table 3: what is the relationship between Cho et al, 2019 report and Bock et al 2021 report which both use HEK-293 cells for study. 

5. In Table 3: what tumor was studied in Timbergen et al report?

6. In Table 3: Cho et al report, there is a typo, "brow".

7. In Table 3: please add "neurons" in Wang report model.

8. Generally, PERM1 is the novel transcription regulator. What would be the future directions for PERM1 study in mitochondrial field and other field.

Author Response

Comment 1. In Figure 1: please use "skeletal muscle" for muscle (n-13), "cardiac muscle" for heart (n=9).

Response 1: Change made as requested.

Comnent 2. In Table 2: please use "cardiac muscle" for heart tissue.

Response 2: Change made as requested

Comment 3. In Table 3: for mice study which is focused on adipose tissues, please add "adipose" in model.

Response 3: Change made as requested

Comment 4. In Table 3: what is the relationship between Cho et al, 2019 report and Bock et al 2021 report which both use HEK-293 cells for study. 

Response 4: Together, the data from Cho et al. (2019) and Bock et al. (2021) suggest that CaMKII and PERM1 may form a multi-protein complex and interact with each other on the outer mitochondrial membrane.

We include the following details in “PERM1 and other tissues” section (Page 18, lines 423-424):

“Moreover, in HEK-293 cells, PERM1 anchors into the outer mitochondrial membrane via its C-terminal domain and binds to CaMKII protein [13, 22], suggesting that CaMKII and PERM1 may be part of the same multi-protein complex and interact with each other via the outer mitochondrial membrane.”

Comment 5. In Table 3: what tumor was studied in Timbergen et al report?

Response 5: Change made as requested

Comment 6. In Table 3: Cho et al report, there is a typo, "brow".

Response 6: Change made as requested

Comment 7. In Table 3: please add "neurons" in Wang report model.

Response 7: Change made as requested

Comment 8. Generally, PERM1 is the novel transcription regulator. What would be the future directions for PERM1 study in mitochondrial field and other field.

Response 8: Please see our detailed responses to Reviewer 1 in addition to our new “Limitations and Future Directions” section on Page 18-19, lines 432-474.

Round 2

Reviewer 1 Report

Comments and Suggestions for Authors

The authors have responded positively to my concerns.

Reviewer 2 Report

Comments and Suggestions for Authors

My questions were answered. No more comments.